# Safety by Design: High-Probability Constrained Contextual Bandits

## Abstract

Multi-Armed Bandit algorithms have emerged as a fundamental framework for numerous recent applications, including reinforcement learning from human feedback (RLHF), optimal dosage determination, experimental design, advertising, recommendation systems, and fairness. Safety constraints are commonly incorporated to address real-world requirements such as preventing private information leakage in large language models, avoiding overdosing scenarios, and protecting vulnerable societal or client groups under optimistically deployed policies. One approach to modeling constrained optimization problems involves introducing two parametric unknown signals: a reward signal and a cost signal. The objective remains maximizing the expected reward while the stage-wise constrained formulation requires a specified statistic of the cost signal to remain within a predefined safety interval. Previous research has developed algorithms ensuring that the expected value of the cost signal remains below a desired threshold, with constraints satisfied with high probability. In this work, we extend these concepts to control the actual realization of the cost signal, ensuring it lies within the safety region with high probability. This advancement opens new directions for applications where hard safety constraints must be satisfied not merely in expectation but with near-certainty. We present an algorithm with accompanying regret bounds, initially for linear reward and cost signals, then generalize to broader function classes by parameterizing our results using the eluder dimension.

## 1 Introduction

Bandit algorithms [24, 4, 15] have been employed across a wide range of domains, including reinforcement learning with human feedback (RLHF) [11, 28] Clinical Trials [17, 16, 5], [26, 7], Recommendation Systems [27], Dynamic Pricing [19], LLMs [8], and Fair Allocation [22] as well as others; see [9] for additional references. In these domains, a learner engages in a sequence of interactions with an unknown environment, striving to both learn about the environment and maximize cumulative reward through its actions. The field has witnessed an increased focus on contextual bandits [12], wherein the learner first observes the present context, that is usually a multidimensional vector, prior to selecting from a potentially unlimited range of actions.

Within this broader framework, constrained bandit algorithms become essential when applications involve resource limitations or safety considerations. In bandit literature, these constraints are based either on past reward data, such as in knapsack bandits [6] and fairness constraints [14], or they involve simultaneous signals of reward and cost, focusing on cost constraints, as discussed in [2, 21, 20]. This approach is beneficial for uses like advertising and drug administration, aligning with the reward/cost model mentioned in [21]. In drug dosage scenarios, there is an efficacy signal (reward) and a toxicity signal (cost). This situation is common in Phase II clinical trials, where clinicians adjust dosages to maximize efficacy while keeping toxicity under a certain threshold $\tau$. Another strategy, introduced by [3], uses a binary reward/cost model aimed at maintaining the average cost signals beneath a set limit.

While cumulative or averaged cost control is useful in some applications, a plethora of applications, particularly in medicine, demand stage-wise constraints—that is, controlling the cost at each time step separately. To address this demand, researchers have studied the well-established "safe linear bandit" problem [13, 20, 18], where at each round $t$, every chosen action generates both a reward signal $r_t \in \mathbb{R}$ and a cost signal $c_t \in \mathbb{R}$. The objective is to maximize the expected cumulative reward while ensuring that the expected value of the cost signal remains below a known safety threshold $\tau$ at every round. This constraint satisfaction is guaranteed with high probability across all sources of randomness in the system. However, in applications such as autonomous driving or drug treatments, satisfying a constraint in expectation does not exclude the possibility of catastrophic outcomes at specific rounds. This raises the necessity of developing algorithms that control the actual value of the cost signal within a safe range at each time step, rather than merely controlling its expected value. With this goal in mind, we attempt to answer the following question.

*Is it possible to design constrained low-regret algorithms*
*such that the constraints are never violated with high probability?*

In this work, we give a positive answer to the previous question by controlling the actual realization of the cost signal, requiring it to remain below the same threshold with probability $\mathbb{P}(c_t \leq \tau) \geq 1 - \delta$, where $\delta > 0$ represents a confidence level provided as input to the algorithm. As in previous works, this constraint is satisfied with high probability with respect to all randomness involved in the process.

To accomplish this goal, we propose a *UCB-like* algorithm named *High Probability Constrained UCB* to meet this demand. Our approach is built upon the *Optimism in the Face of Uncertainty* (OFU) principle [4] and eliminates the need for prior knowledge of an initial safe action, offering a notable advantage over current techniques. It functions effectively in both adversarial and stochastic scenarios, depending solely on the standard assumption of sub-Gaussian noise distributions. Utilizing this assumption, we design a constraint event that, with high probability, ensures the cost signal does not exceed the desired threshold in any round.

We establish that our algorithm attains a *T*-round regret bound of the order $\tilde{\mathcal{O}}(d\sqrt{T})$. Furthermore, we showcase the practical success of our method through computational experiments. We also broaden our findings to scenarios with *non-linear* reward and cost functions by framing our analysis using the *eluder dimension*, a complexity metric for function classes.

## 2 Problem Formulation

**Notation.** We adopt the following notation throughout the paper. We denote by $\langle x, y \rangle = x^\top y$ and $\langle x, y \rangle_{\mathbf{A}} = x^\top \mathbf{A} y$, for a positive definite matrix $\mathbf{A} \in \mathbb{R}^{d \times d}$, the inner-product and weighted inner-product of vectors $x, y \in \mathbb{R}^d$. Similarly, we denote by $\|x\| = \sqrt{x^\top x}$ and $\|x\|_{\mathbf{A}} = \sqrt{x^\top \mathbf{A} x}$, the $\ell_2$ and weighted $\ell_2$ norms of vector $x \in \mathbb{R}^d$. We denote the indicator function as $\mathbf{1}\{\cdot\}$. We use upper-case letters for random variables (e.g., $X$), and their corresponding lower-case letters for a particular instantiation of that random variable (e.g., $X = x$). The set $\{1, \ldots, T\}$ is denoted by $[T]$. Finally, we use $\tilde{\mathcal{O}}$ for the big-$\mathcal{O}$ notation up to logarithmic factors.

Inspired by bandit algorithms designed for RLHF and the adaptive dosage allocation problem, we adopt the following formulation for the action set and the reward and cost signals. At each iteration $t \in [T]$, the learner observes a $d$-dimensional context vector $X_t \in \mathbb{R}^d$, which may represent medical test results or a language model (LM) embedding of a prompt-answer pair. We impose no distributional assumptions on the context $X_t$; it may be stochastically generated or adversarial. The learner then selects a scalar action $\alpha_t \in [0, 1]$, and the environment generates the reward and cost signals as follows: the reward signal is $R_t := \alpha_t \cdot (r(X_t) + \xi_t^r)$ and the cost signal is $C_t := \alpha_t \cdot (c(X_t) + \xi_t^c)$, where $\xi_t^r, \xi_t^c$ denote subgaussian noise terms, and $r(X_t), c(X_t)$ measure the importance or significance of the context to the reward and cost mechanisms, respectively. Initially, we model $r(X_t), c(X_t)$ as linear functions parameterized by unknown vectors $\theta^\star$ and $\mu^\star$, respectively. We subsequently generalize our results by requiring only that $r(X_t), c(X_t)$ be bounded.

We now provide motivation for our choice of protocol and the reward and cost function formulation. Drawing inspiration from optimal dosage applications, the context $X_t$ describes the medical condition of a patient, the reward function $r(X_t)$ measures the therapeutic effect of a drug on the patient's current state, and $c(X_t)$ measures the drug's side effects. In this setting, the action $\alpha_t$ denotes the dosage assigned to the patient. If $r(X_t) >> 0$, then the drug is beneficial to the patient, and we should assign the maximum possible dosage without overdosing the patient. In advertising applications, each

---

**Safe Bandit protocol**

---

**Input:** Horizon $T$

For rounds $t = 1, 2, \ldots, n$:

1. 👤 $\xleftarrow{X_t \in \mathbb{R}^d}$ 🌐 % context
2. 👤 $\xrightarrow{\alpha_t \in [0,1]}$ 🌐 % action selection
3. 👤 $\xleftarrow{R_t, C_t \in \mathbb{R}}$ 👤 % reward, cost signals

---

advertisement has both positive and negative impacts on an audience, and $\alpha_t$ denotes the duration for which we display the advertisement. Finally, in RLHF, the contexts can represent the embedding of prompt-candidate answer pairs. The reward function can measure the satisfaction a user derives from a given answer to their prompt, while the cost function can be implemented as an LLM fine-tuned on safety parameters that evaluates how acceptable or safe the provided answer is. The actions $\alpha_t$ determine the level of reasoning effort devoted to each prompt, allowing the system to limit computational investment in potentially malicious or adversarial queries.

Before proceeding with our analysis and results, we first outline the standard assumptions for the model. These assumptions are well-established in the literature on contextual bandits with constraints.

**Assumption 2.1** (Sub-Gaussian noise). *For all $t \in [T]$, the reward and cost noise random variables $\xi_t^r$ and $\xi_t^c$ are conditionally Sub-Gaussian, i.e., for all $\alpha \in \mathbb{R}$, and the Sub-Gaussianity constants $\gamma_r, \gamma_c > 0$;*

$$\mathbb{E}[\xi_t^r \mid \mathcal{H}_{t-1}] = 0, \quad \mathbb{E}[\exp(\alpha \xi_t^r) \mid \mathcal{H}_{t-1}] \leq \exp(\alpha^2 \gamma_r^2 / 2),$$

$$\mathbb{E}[\xi_t^c \mid \mathcal{H}_{t-1}] = 0, \quad \mathbb{E}[\exp(\alpha \xi_t^c) \mid \mathcal{H}_{t-1}] \leq \exp(\alpha^2 \gamma_c^2 / 2),$$

*where $\mathcal{H}_t$ is the filtration that includes all the events $(X_{1:t+1} R_{1:t}, C_{1:t}, \xi_{1:t}^r, \xi_{1:t}^c)$ until the end of round $t$.*

**Assumption 2.2** (bounded parameters). *There is a known constant $S > 0$, such that $\|\theta_*\| \leq S$ and $\|\mu_*\| \leq S$.*[1]

**Assumption 2.3** (bounded contexts). *The $\ell_2$-norm of all contexts are bounded by $L > 0$, i.e.,*

$$\max_{t \in [T]} \|X_t\| \leq L.$$

**Assumption 2.4** (Positive toxicity threshold). *The toxicity constraint in order to be meaningful must satisfy that $\tau > 0$.*

We observe that our analysis does not require knowledge of an initial safe action, unlike in [20] or any assumption about the initial decision set like in [18]. However, we believe that in their analysis, this assumption can be relaxed as the vector $\mu^*$ is bounded, and any $X_t$ from their decision set satisfying $\|X_t\| \leq \frac{\tau}{S}$ can serve as an initial safe action. They mention this possibility in their related works. This follows from the inequality $\langle X_t, \mu^* \rangle \leq \|X_t\| \|\mu^*\| \leq \frac{\tau}{S} \cdot S = \tau$.

In each round $t$, the agent is constrained to select an action $\alpha_t$ such that $\alpha_t \left( \langle X_t, \mu^* \rangle + \gamma_c \sqrt{2 \log\left(\frac{1}{\delta}\right)} \right) \leq \tau$. We demonstrate in Section 3 that when this constraint is satisfied, it ensures $\mathbb{P}_{\xi_t^c} (C_t \leq \tau \mid \mathcal{H}_t) \geq 1 - \delta$. We define the set of feasible dosages as

$$\mathcal{A}_t^f = \left\{ \alpha \in [0,1] : \alpha \left( \langle X_t, \mu^* \rangle + \gamma_c \sqrt{2 \log\left(\frac{1}{\delta}\right)} \right) \leq \tau \right\}.$$

Because $\mu^*$ is not known, this set is originally unknown, which requires us to estimate it.

Maximizing the expected reward over $T$ rounds is equivalent to minimizing the expected $T$-round constrained pseudo-regret, defined as

$$\mathcal{R}_\mathcal{C}(T) = \sum_{t=1}^{T} (\alpha_t^* - \alpha_t) \langle X_t, \theta^* \rangle, \tag{1}$$

---

[1]The choice of the same upper-bound $S$ for both $\theta_*$ and $\mu_*$ is just for simplicity and convenience.

where $\alpha_t^*$ represents the optimal feasible action for round $t$, i.e., $\alpha_t^* \in \arg\max_{\alpha \in \mathcal{A}_t^f} \alpha \langle X_t, \theta^* \rangle$. On its side, $\alpha_t$ is the action selected by the learner in round $t$, which is chosen from the set of feasible actions available in that round, i.e., $\alpha_t \in \mathcal{A}_t^f$, with high probability subject to $\xi_{1:t-1}^c, C_{1:t-1}$.

## 3  Constraint formulation

The learner's goal is to maximize the cumulative reward over $T$ rounds, i.e., $\sum_{t=1}^T \alpha_t \langle X_t, \theta^* \rangle$, while ensuring that the realized toxicity remains below a known threshold with high probability. In clinical trials terminology, this problem is referred to as a Phase II trial, as the toxicity threshold $\tau$ is considered known in advance. Our algorithm takes as input a confidence level $\delta$ to control the realization of the noise. To model the requirement of controlling the cost realization with high probability, we impose a nonlinear constraint involving $\delta$. It should be noted that if we know the exact distribution of the noise (i.e. Normal), this problem can be solved exactly without introducing this constraint by using a similar algorithm.

We formulate the following constraint:

**Lemma 1.** *When the selected dosage $\alpha_t$ satisfies $\alpha_t \left( \langle X_t, \mu^* \rangle + \gamma_c \sqrt{2 \log\left(\frac{1}{\delta}\right)} \right) \leq \tau$ then it holds that* $\Pr(C_t \leq \tau \mid \mathcal{H}_t) \geq 1 - \delta$.

The proof is provided in Appendix A.1 and it is a direct application of a concentration bound for Sub-Gaussian random variables (see [15], chapter 5, or [25] chapter 2).

We note that, given the distribution of the noise at round $t$, $\xi_t^c$, it holds that $\Pr(C_t \leq \tau \mid \mathcal{H}_t) \geq 1 - \delta$. The constraint

$$\alpha_t \left( \langle X_t, \mu^* \rangle + \gamma_c \sqrt{2 \log\left(\frac{1}{\delta}\right)} \right) \leq \tau.$$

is thus satisfied with high probability with respect to $\mathcal{H}_{t-1}$.

Since $\tau > 0$, we show that an initial safe interval for choosing $\alpha_1$ is

$$\left[ 0, \min\left( 1, \frac{\tau}{\gamma_c \sqrt{2 \log\left(\frac{1}{\delta}\right)} + LS} \right) \right].$$

To begin with, if

$$\langle X_t, \mu^* \rangle + \gamma_c \sqrt{2 \log\left(\frac{1}{\delta}\right)} \leq 0,$$

then $\mathcal{A}_0^f = [0, 1]$. Otherwise, $\alpha_0$ can range from 0 up to

$$\min\left( 1, \frac{\tau}{\gamma_c \sqrt{2 \log\left(\frac{1}{\delta}\right)} + LS} \right),$$

since $\langle X_t, \mu^* \rangle \leq L \cdot S$ by the Cauchy–Schwarz inequality.

## 4  Algorithm

We aim for our algorithm to leverage the fundamental principle of Optimism in the Face of Uncertainty (OFU). Additionally, we need to make robust choices to ensure that the constraint is satisfied with high probability. To achieve this, we intend to be optimistic in our estimates for the reward signal and pessimistic for the cost.

In each non-zero dose round, we construct two least squares estimators: one for $\theta^*$ and one for $\mu^*$. For a given regularization parameter $\lambda > 0$, the regularized covariance matrix at round $t$ is defined as:

$$\Sigma_t = \lambda I + \sum_{s=1}^{t-1} X_s X_s^\top. \tag{2}$$

Using Equation (2), we define the regularized least squares estimators $\hat{\theta}_t$ and $\hat{\mu}_t$.

$$\hat{\theta}_t = \Sigma_t^{-1} \sum_{s : \alpha_s \neq 0} \alpha_s^{-1} R_s X_s \quad \hat{\mu}_t = \Sigma_t^{-1} \sum_{s : \alpha_s \neq 0} \alpha_s^{-1} C_s X_s \tag{3}$$

**Algorithm 1** High Probability Constrained UCB

**Require:** Constraint threshold $\tau \geq 0$, confidence parameter $\delta$, sub-Gaussianity constants $\gamma_c, \gamma_r$

1: $\alpha_0 \leftarrow \min\left\{1, \dfrac{\tau}{\gamma_c\sqrt{2\log\left(\frac{1}{\delta}\right)}+L\cdot S}\right\}$

2: **for** $t = 1, 2, \ldots, T$ **do**

3:      Compute $\hat{\mu}$ according to (3)

4:      Use $\hat{\mu}$ to compute the estimated feasible set $\hat{\mathcal{A}}_t^f$ using (7)

5:      Compute $\hat{\theta}_t$ using (5)

6:      Compute action $\alpha_t = \arg\max_{\alpha \in \hat{\mathcal{A}}_t^f} \alpha \langle X_t, \tilde{\theta}_t \rangle$

7:      Take action $\alpha_t$ and if $\alpha_t \neq 0$ store the reward and cost signals $(R_t, C_t)$

8: **end for**

---

We note that we use only the contexts $X_t$ and the corresponding realizations of the reward and cost signals $R_t$ and $C_t$ for the rounds in which we assigned a non-zero action. In rounds where we selected an action equal to zero, we did not receive feedback about the dosage effect; that is, $R_t = C_t = 0$, given the way our model is constructed.

To design a UCB-like algorithm, we need to define high-probability confidence sets centered at our estimators $\hat{\theta}_t$ and $\hat{\mu}_t$. These confidence sets will enable us to derive upper bounds on the distances between our estimators and the unknown vectors $\theta^*$ and $\mu^*$. To construct the desired confidence intervals, we will use the following fundamental theorem.

**Theorem 1** (Theorem 2 in [1]). *For a fixed $\delta \in (0, 1)$ and $\forall t \in [T]$;*

$$\beta_t^r(\delta, d) = \gamma_r\sqrt{d\log\left(\frac{1 + (t-1)L^2/\lambda}{\delta}\right)} + \sqrt{\lambda}S,$$

$$\beta_t^c(\delta, d) = \gamma_c\sqrt{d\log\left(\frac{1 + (t-1)L^2/\lambda}{\delta}\right)} + \sqrt{\lambda}S,$$

*it holds with probability at least $1 - \delta$ that*

$$\|\widehat{\theta}_t - \theta_*\|_{\Sigma_t} \leq \beta_t^r(\delta, d), \qquad \|\widehat{\mu}_t - \mu_*\|_{\Sigma_t} \leq \beta_t^c(\delta, d).$$

We will make use of Theorem 1 to define the following confidence sets (ellipsoids):

$$\begin{aligned}
\mathcal{C}_t^r &= \{\theta \in \mathbb{R}^d : \|\theta - \widehat{\theta}_t\|_{\Sigma_t} \leq \beta_t^r(\delta, d)\}, \\
\mathcal{C}_t^c &= \{\mu \in \mathbb{R}^d : \|\mu - \widehat{\mu}_t\|_{\Sigma_t} \leq \beta_t^c(\delta, d)\},
\end{aligned} \tag{4}$$

Theorem 1 suggests that $\theta^* \in \mathcal{C}_t^r$ and $\mu^* \in \mathcal{C}_t^c(\alpha_c)$, each with probability at least $1 - 2\delta$. We will use these confidence intervals to create our estimators for $\theta^*$ and $\mu^*$.

We aim to be optimistic in our estimate of $\theta^*$ by selecting

$$\tilde{\theta}_t = \arg\max_{\theta \in \mathcal{C}_t^r} \langle X_t, \theta \rangle, \tag{5}$$

and pessimistic about $\mu^*$ by choosing $\tilde{\mu}_t$ that minimizes the volume of the estimated feasible set. In our case, the feasible set is a continuous sub-interval of $[0, 1]$, so its measure is simply its length. Before describing the algorithm, we will first define the confidence ellipsoids and the Least Squares Estimators for $\theta^*$ and $\mu^*$.

The computation of the estimated feasible set $\hat{\mathcal{A}}_t^f$ is performed in two steps. First, we estimate the unknown cost vector $\mu^*$ using a least squares estimator. This procedure yields a confidence ellipsoid that contains $\mu^*$ with high probability. Among all $\mu$ within this ellipsoid, we select the one that minimizes the length of the interval of feasible values for $\alpha_t$.

## 4.1 Choice of $\hat{\mu}$

As previously discussed, we aim to choose our estimate pessimistically to minimize the length of $\hat{\mathcal{A}}_t^f$. By definition, $\hat{\mathcal{A}}_t^f$ is given by

$$\hat{\mathcal{A}}_t^f = \left\{\alpha \in [0,1] : \alpha \left(\langle X_t, \tilde{\mu}\rangle + \gamma_c \sqrt{2\log\left(\tfrac{1}{\delta}\right)}\right) \leq \tau\right\}.$$

Since $\tau > 0$, we first need to check the sign of $\langle X_t, \tilde{\mu}\rangle + \gamma_c \sqrt{2\log\left(\tfrac{1}{\delta}\right)}$. If this expression is negative for all $\mu \in \mathcal{C}_\mu^t$, then we set $\hat{\mathcal{A}}_t^f = [0,1]$. However, if there exists a $\mu \in \mathcal{C}_\mu^t$ such that this expression is positive, we select the $\mu$ that minimizes the maximum feasible $\alpha_t$. This approach can be summarized in the following convex program, where $\hat{\mu}$ is the least squares estimate of $\mu$.

$$\begin{aligned}
\max_{\mu} \quad & \langle X_t, \mu\rangle \\
\text{subject to} \quad & \|\mu - \hat{\mu}\|_{\Sigma_t} \leqslant \beta_t^{r\,2}, \\
& \langle X_t, \mu\rangle + \gamma_c \sqrt{2\log\left(\frac{1}{\delta}\right)} \geqslant 0
\end{aligned} \tag{6}$$

Let $\mathcal{K}_\mu(t) = \{\mu \in \mathbb{R}^d : \|\mu - \hat{\mu}\|_{\Sigma_t} \leqslant \beta_t^{r\,2}, \langle X_t, \mu\rangle + \gamma_c \sqrt{2\log\left(\tfrac{1}{\delta}\right)} \geqslant 0\}$ the be set of feasible solutions of the convex program 6. If $\mathcal{K}_\mu(t) \neq \emptyset$, then let $\tilde{\mu} \in \arg\max_{\mu \in \mathcal{K}_\mu(t)}\{\langle X_t, \mu\rangle\}$.

$$\hat{\mathcal{A}}_t^f = \begin{cases} [0,1] & , \text{if } \mathcal{K}_\mu(t) = \emptyset \\ \left[0, \dfrac{\tau}{\langle X_t, \tilde{\mu}\rangle + \gamma_c \sqrt{2\log\left(\tfrac{1}{\delta}\right)}}\right] & , \text{if } \mathcal{K}_\mu(t) \neq \emptyset \end{cases} \tag{7}$$

## 5 Regret Analysis

The objective of the agent is to minimize the *expected T-round (constrained) (pseudo)-regret*, i.e.,

$$\mathcal{R}_\mathcal{C}(T) = \sum_{t=1}^{T} r^*(X_t) - r(X_t),$$

where

$$r^*(X_t) = \max_{\alpha \in \mathcal{A}_t^f} \alpha\langle X_t, \theta^*\rangle,$$

$$r(X_t) = \max_{\alpha \in \hat{\mathcal{A}}_t^f} \alpha\langle X_t, \theta^*\rangle.$$

We see that the choice of $\alpha$ depends on the sign of $\langle X_t, \theta^*\rangle$. If this inner product is positive we choose the largest feasible value and otherwise the lowest feasible one.

$$\begin{aligned}
\mathcal{R}_\mathcal{C}(T) &= \sum_{t=1}^{T} \max_{\alpha \in \mathcal{A}_t^f} (\alpha\langle X_t, \theta^*\rangle) - \alpha_t\langle X_t, \theta^*\rangle \\
&= \sum_{t=1}^{T} (\alpha_t^* - \alpha_t)\langle X_t, \theta^*\rangle.
\end{aligned} \tag{8}$$

We will use a decomposition of the regret similar to standard ones in the *Linear Bandits under constraints* literature, ([2],[21],[20]). We define as

$$\tilde{\alpha}_t = \arg\max_{\alpha \in \mathcal{A}_t^f}\{\alpha\langle X_t, \hat{\theta}_t\rangle\}. \tag{9}$$

Using the above definition we decompose the regret as follows.

$$
\begin{aligned}
\mathcal{R}_{\mathcal{C}}(T) &= \sum_{t=1}^{T} (\alpha_t^* - \alpha_t)\langle X_t, \theta^* \rangle \\
&= \sum_{t=1}^{T} \underbrace{(\alpha_t^* - \tilde{\alpha}_t)\langle X_t, \theta^* \rangle}_{\text{Term 1: Cost for approximating } \theta^*} \\
&\quad + \sum_{t=1}^{T} \underbrace{(\tilde{\alpha}_t - \alpha_t)\langle X_t, \theta^* \rangle}_{\text{Term 2: Cost for approximating } \mu^*} .
\end{aligned}
\tag{10}
$$

## 5.1 Analysis of the Regret

**Lemma 2.** *The first term in the regret decomposition can be bounded as follows:*

$$
(\alpha_t^* - \tilde{\alpha}_t)\langle X_t, \theta^* \rangle \leq \tilde{\alpha}_t \langle X_t, \tilde{\theta} - \theta^* \rangle.
$$

The proof is in Appendix A.2. We note that this is the standard bound in the Linear Bandits literature as first proved in the classical work of [1]. It remains to bound the second term.

**Lemma 3.** *The second term in the regret decomposition can be bounded as follows:*

$$
(\tilde{\alpha}_t - \alpha_t)\langle X_t, \theta^* \rangle \leq L \cdot S \cdot \frac{\langle X_t, \tilde{\mu} - \mu^* \rangle}{\tau}.
$$

The proof is in Appendix A.3.

By using the lemmas 2, 3 combining with the regret decomposition (equation 10) we can bound the regret as following. The bound is conditioned on the following event that holds with probability at least $1 - 2\delta'$.

$$
\mathcal{E} := \left\{ \|\tilde{\theta}_t - \widehat{\theta}_t\|_{\Sigma_t} \leq \beta_t(\delta', d) \wedge \|\tilde{\mu}_t - \widehat{\mu}_t\|_{\Sigma_t} \leq \beta_t(\delta', d) \right\}.
\tag{11}
$$

It is important to mention that $\delta'$ is not necessary equal to $\delta$. The first one is the probability that the regret bounds holds and the second one the probability that the realization of the noise of the cost stays below the threshold.

**Theorem 2.** *With probability at least one $1 - \delta'$ the regret of the High Probability Constrained UCB algorithm can be bounded by $\mathcal{O}\left( \frac{L \cdot S}{\tau} \cdot \beta_T(\delta', d)\sqrt{2Td\log\left(1 + \frac{TL^2}{\lambda}\right)} \right)$.*

The proof is in A.4.

# 6 Non-linear rewards and costs

Instead of modeling the reward and the cost signal as linear functions in term of the unknown parameters $\theta$ and $\mu$ we can use more general functions and express our results in terms of the *Eluder dimension* as defined in [23].

We denote the set of feasible actions in round $t$ as $\mathcal{A}_t(X_t) = \{\alpha \in [0,1] \mid \alpha\left(\mu_*(X_t) + \gamma_c\sqrt{2\log(\frac{1}{\delta})}\right) \leq \tau\}$. The agent selects and action $\alpha_t \in \mathcal{A}_t(X_t)$. Now the reward and the cost signal take the following form.

$$
R_t = \alpha_t\theta_*(X_t) + \alpha_t\xi_t^r, \quad C_t = \alpha_t\mu_*(X_t) + \alpha_t\xi_t^c,
$$

where $\theta_*(\cdot) \in \mathcal{G}_r$ and $\mu_*(\cdot) \in \mathcal{G}_c$ are the mean reward and cost function respectively that belong to the known function classes $\mathcal{G}_r, \mathcal{G}_c$. We will assume that $\theta_*(\cdot), \mu_*(\cdot)$ take values in $[-1, 1]$, relaxing the standard assumption made that the non-linear functions take values in $[0, 1]$. We show that the important property is that the non-linear functions remain bounded. We also assume that the reward and the cost signals are bounded, i.e. lie in $[-1, 1]$. For the noise signals $\xi_t^r, \xi_t^c$ we assume that they are conditionally sub-Gaussian. Moreover, we use the definition of the width of a subset $\tilde{\mathcal{F}} \subset \mathcal{F}$ at a context $X \in \mathcal{A}$ by

$$w_{\tilde{\mathcal{F}}}(X) = \sup_{\underline{f}, \overline{f} \in \tilde{\mathcal{F}}} \left( \overline{f}(X) - \underline{f}(X) \right). \tag{12}$$

In the new terminology, the $T$ period regret is written as

$$\mathcal{R}(T, \pi) = \sum_{t=1}^{T} \left[ \alpha_t^* \theta_*(X_t) - \alpha_t \theta_*(X_t) \right].$$

First we define the dataset $\mathcal{D}_t = \{(X_s, R_s, C_s)\}_{s=1}^{t-1}$ for $s$ such that $A_s \neq 0$, that is the dataset of observed information up to the beginning of round $t$, and $\|f\|_{\mathcal{D}_t} = \sqrt{\sum_{x \in \mathcal{D}_t} f^2(x)}$ the norm induced by the dataset for any function $f : \mathcal{A}_t \to \mathbb{R}$.

In every round we define the confidence ellipsoids as follows

$$\mathcal{C}_t^r(\delta) = \{\theta \in \mathcal{G}_r : \left\| \theta - \hat{\theta} \right\|_{\mathcal{D}_t} \leq \rho_r(t, \delta/2)\}$$

$$\mathcal{C}_t^c(\delta) = \{\theta \in \mathcal{G}_c : \left\| \theta - \hat{\theta} \right\|_{\mathcal{D}_t} \leq \rho_c(t, \delta/2)\}$$

Using these confidence intervals we compute the actions of the algorithm as follows. To compute the feasible dosages, first we solve the following Non-Linear program.

$$\begin{aligned} \max_{\mu} \quad & \mu(X_t) \\ \text{subject to} \quad & \|\mu(X_t) - \hat{\mu}(X_t)\|_{\mathcal{D}_t} \leqslant \beta_t^2, \\ & \mu(X_t) + \gamma_c \sqrt{2 \log\left(\frac{1}{\delta}\right)} \geqslant 0 \end{aligned} \tag{13}$$

Then if there is no feasible solution in the above optimization problem we select $\hat{A}_t = [0, 1]$ otherwise, let say $\mathcal{K}(\hat{\mu}_t)$ its solution, then $\hat{\mathcal{A}}_t^f = [0, \frac{\tau}{\mathcal{K}(\hat{\mu}_t) + \gamma_c \sqrt{2 \log\left(\frac{1}{\delta}\right)}}]$ as before.

Our estimate for $\theta$ is $\tilde{\theta}(X_t) = \max_{\theta \in \mathcal{C}_t^r(\delta')} \theta(X_t)$.

---

**Algorithm 2** Non-Linear High Probability Constrained UCB

1: **Input:** Constraint threshold $\tau \geq 0$;   Confidence parameter $\delta$;   Sub-Gaussianity constant $\gamma_c$
2: $\alpha_0 \leftarrow \min\{1, \frac{\tau}{\gamma_c \sqrt{2 \log\left(\frac{1}{\delta}\right)} + \max_X \mu_*(X)}\}$
3: **for** $t = 1, 2, \cdots, T$ **do**
4:     Compute $\hat{\mu}, \hat{\theta}$ by using Least Squares Estimators
5:     Construct the $\hat{\mathcal{A}}_t^f, \tilde{\theta}(X_t)$
6:     Compute action $\alpha_t = \arg\max_{\alpha \in \hat{\mathcal{A}}_t^f} \alpha \tilde{\theta}(X_t)$
7:     Take action $\alpha_t$ and if $\alpha_t \neq 0$ store the reward and the cost signals $(R_t, C_t)$
8: **end for**

---

We want to apply the same regret decomposition as before. First, we define analogously $\hat{\mathcal{A}}_t(X_t) = \{\alpha \in [0, 1] \mid \alpha \left( \mu_*(X_t) + \gamma_c \sqrt{2 \log(\frac{1}{\delta})} \right) \leq \tau\}$. We also define

$$\alpha_t^* \in \arg\max_{\alpha \in \mathcal{A}_t} \theta_*(\alpha).$$

$$\alpha_t \in \arg\max_{\alpha \in \hat{\mathcal{A}}_t} \sup_{\theta \in \mathcal{G}_r} \theta(\alpha).$$

$$\tilde{\alpha}_t \in \arg\max_{\alpha \in \mathcal{A}_t} \sup_{\theta \in \mathcal{G}_r} \theta(\alpha).$$

As in **Proposition 1** in [23] our goal is to bound the regret using $w_{\tilde{\mathcal{F}}}(X_t)$. First we apply the same decomposition to express the regret in terms of the cost due to the lack of knowledge of $\theta_*$ and $\mu_*$.

$$\mathcal{R}(T,\pi) = \sum_{t=1}^{T} [\alpha^* \theta_*(X_t) - \tilde{\alpha}_t \theta_*(X_t)]$$
$$+ \sum_{t=1}^{T} [\tilde{\alpha}_t \theta_*(X_t) - \alpha_t \theta_*(X_t)].$$

The first sum can be bounded in a similar way to **proof A** in the appendix of [23]. The second sum measures the regret the algorithm suffers from the lack of knowledge of $\mu$. Then we can bound in terms of $w_{\tilde{\mathcal{F}}_\mu}$ the same way as before.

**Lemma 4.** $\alpha_t^* \theta_*(X_t) - \tilde{\alpha}_t \theta_*(X_t) \leq w_{\mathcal{G}_r}(X_t) + 2\mathbf{1}\{(\theta_* \notin \mathcal{G}_r)\}.$

The proof is in B.1.

For the remaining part, we need to bound $|\tilde{\alpha}_t - \alpha_t|$ in terms of $\mu_*$. We will follow a similar proof as in the case of the inner product function.

**Lemma 5.** $|\tilde{\alpha}_t - \alpha_t| \leq w_{\mathcal{G}_c}(X_t)/\tau.$

The proof is similar to the linear case and it is provided in B.2.

Now that we have bound the regret in terms of the width of the set that the non-linear functions belong we can translate our results to bound for the regret. First, as in the linear model case, we define the reward and the cost set confidence radii as in [20].

$$\rho_r(t,\delta') = 512 \log\left(\frac{24|\mathcal{G}_r|\log(2t)}{\delta}\right),$$
$$\rho_c(t,\delta') = 512 \log\left(\frac{24|\mathcal{G}_c|\log(2t)}{\delta}\right).$$

We also use the following notation $d_{eluder}^r = d_{eluder}(\mathcal{G}_r, 1/T)$ and $d_{eluder}^c = d_{eluder}(\mathcal{G}_c, 1/T)$. The algorithm is similar to that one in the linear case. For the regret bound, like [20], we use the Lemma 3 in [10], by setting $P = 1$.

**Theorem 3.** *With probability at least* $1-\delta'$, *the regret of the Non-Linear High Probability Constrained UCB satisfies*

$$\mathcal{R}(T) = \mathcal{O}(\sqrt{T d_{eluder}^r \rho_r(T,\delta'/2)} +$$
$$1/\tau \sqrt{T d_{eluder}^c \rho_c(T,\delta'/2)} +$$
$$d_{eluder}^r + \frac{d_{eluder}^c}{\tau}).$$

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

# A Appendix

## A.1 Constraint formulation

We assume that the cost noise is conditionally sub-Gaussian with a known constant $\gamma_c$. Under this assumption, the random variable $\frac{C_t - \alpha_t \langle X_t, \mu \rangle}{\alpha_t}$ is $\gamma_c$ sub-Gaussian. We will define a constraint event such that, when satisfied, the cost signal remains below the threshold with high probability. Now, we can analyze the cost using the following theorem.

**Theorem 4.** *[Sub Gaussian concentration bounds - Theorem 5.3 [15]] If $X$ is $\gamma_c$-subgaussian, then for any $\epsilon > 0$,*

$$\Pr(X \geq \epsilon) \leq \exp(-\frac{\epsilon^2}{2\gamma_c^2})$$

Using the above property of the cost noise and the theorem we derive that

$$\Pr(C_t \geq \tau \mid \mathcal{H}_t) = \Pr\left( \frac{C_t - \alpha_t \langle X_t, \mu^* \rangle}{\alpha_t} \geq \frac{\tau - \alpha_t \langle X_t, \mu^* \rangle}{\alpha_t} \,\middle|\, \mathcal{H}_t \right) \tag{14}$$

$$\leq \exp\left( -\frac{\left( \frac{\tau - \alpha_t \langle X_t, \mu^* \rangle}{\alpha_t} \right)^2}{2\gamma_c^2} \right) \tag{15}$$

By requiring the right-hand side to be less than or equal to $\delta$, we derive:

$$\exp\left( -\frac{\left( \frac{\tau - \alpha_t \langle X_t, \mu^* \rangle}{\alpha_t} \right)^2}{2\gamma_c^2} \right) \leq \delta$$

$$\frac{\left( \frac{\tau - \alpha_t \langle X_t, \mu^* \rangle}{\alpha_t} \right)^2}{2\gamma_c^2} \geq \log\left( \frac{1}{\delta} \right) \tag{16}$$

$$\frac{\tau - \alpha_t \langle X_t, \mu^* \rangle}{\alpha_t} \geq \gamma_c \sqrt{2 \log\left( \frac{1}{\delta} \right)}$$

$$\tau \geq \alpha_t \left( \langle X_t, \mu^* \rangle + \gamma_c \sqrt{2 \log\left( \frac{1}{\delta} \right)} \right)$$

## A.2 Analyzing the cost for approximating $\theta$

The first term need to be bounded is $\sum_{t=1}^{T} (\alpha_t^* - \tilde{\alpha}_t) \langle X_t, \theta^* \rangle$. In order to bound this term we will follow a standard procedure in Linear Bandits. Initially, we will bound the term $\alpha_t^* \langle X_t, \theta^* \rangle$. With

probability at least $1 - \delta$ it holds $\theta^* \in \mathcal{C}_t^\theta , \forall t \in [T]$.

$$\alpha_t^* \langle X_t, \theta^* \rangle \leq \max_{\theta \in \mathcal{C}_t^\theta} \{\alpha_t^* \langle X_t, \theta \rangle\}$$

$$\leq \max_{\alpha \in \mathcal{A}_t^f} \max_{\theta \in \mathcal{C}_t^\theta} \{\alpha \langle X_t, \theta \rangle\}$$

$$= \max_{\alpha \in \mathcal{A}_t^f} \{\alpha \langle X_t, \tilde{\theta} \rangle\}$$

$$= \tilde{\alpha}_t \langle X_t, \tilde{\theta} \rangle$$

Using the above it holds that

$$(\alpha_t^* - \tilde{\alpha}_t)\langle X_t, \theta^* \rangle \leq \tilde{\alpha}_t \langle X_t, \tilde{\theta} - \theta^* \rangle$$

## A.3 Analyzing the cost for approximating $\mu^*$

We can bound the second term using the Cauchy-Schwarz inequality as follows:

$$(\tilde{\alpha}_t - \alpha_t)\langle X_t, \theta^* \rangle \leq |\tilde{\alpha}_t - \alpha_t| \cdot LS.$$

It remains to bound $|\tilde{\alpha}_t - \alpha_t|$. First, we remind the definitions of $\tilde{\alpha}_t$ and $\alpha_t$:

$$\tilde{\alpha}_t = \arg\max_{\alpha \in \mathcal{A}_t^f} \{\alpha \langle X_t, \hat{\theta}_t \rangle\},$$

$$\alpha_t = \arg\max_{\alpha \in \hat{\mathcal{A}}_t^f} \{\alpha \langle X_t, \hat{\theta}_t \rangle\}.$$

We observe that both the choice of $\tilde{\alpha}_t$ and the choice of $\alpha_t$ depend on the sign of the inner product $\langle X_t, \hat{\theta}_t \rangle$. If $\langle X_t, \hat{\theta}_t \rangle \geq 0$, then $\tilde{\alpha}_t$ equals the maximum element of the set $\mathcal{A}_t^f$. Similarly, $\alpha_t$ equals the maximum of the set $\hat{\mathcal{A}}_t^f$ when $\langle X_t, \hat{\theta}_t \rangle \geq 0$. On the other side, when $\langle X_t, \hat{\theta}_t \rangle < 0$, both $\tilde{\alpha}_t$ and $\alpha_t$ are zero.

We will write down again the sets $\mathcal{A}_t^f$ and $\hat{\mathcal{A}}_t^f$ to see the possible values for $(\tilde{\alpha}_t, \alpha_t)$:

$$\mathcal{A}_t^f = \left\{ \alpha \in [0,1] : \left( \langle X_t, \mu^* \rangle + \gamma_c \left( \sqrt{2\log\left(\tfrac{1}{\delta}\right)} \right) \right) \alpha \leq \tau \right\},$$

$$\hat{\mathcal{A}}_t^f = \left\{ \alpha \in [0,1] : \left( \langle X_t, \tilde{\mu} \rangle + \gamma_c \left( \sqrt{2\log\left(\tfrac{1}{\delta}\right)} \right) \right) \alpha \leq \tau \right\}.$$

Our estimator $\tilde{\mu}$ for $\mu^*$ is a pessimistic one. Among all possible choices for $\tilde{\mu}$, in order to be robust, we will choose $\tilde{\mu}$ such that $\hat{\mathcal{A}}_t^f$ has the smallest possible length.

Having that in mind, we have the four following scenarios for $(\tilde{\alpha}_t, \alpha_t)$:

1. $(1, 1)$

2. $\left( 1, \min \left( 1, \dfrac{\tau}{\gamma_c \left( \sqrt{2\log\left(\tfrac{1}{\delta}\right)} \right) + \langle X_t, \tilde{\mu} \rangle} \right) \right)$

3. $\left( \min \left( 1, \dfrac{\tau}{\gamma_c \left( \sqrt{2\log\left(\tfrac{1}{\delta}\right)} \right) + \langle X_t, \mu^* \rangle} \right), 1 \right)$

4. $\left( \min \left( 1, \dfrac{\tau}{\gamma_c \left( \sqrt{2\log\left(\tfrac{1}{\delta}\right)} \right) + \langle X_t, \mu^* \rangle} \right), \min \left( 1, \dfrac{\tau}{\gamma_c \left( \sqrt{2\log\left(\tfrac{1}{\delta}\right)} \right) + \langle X_t, \tilde{\mu} \rangle} \right) \right)$

For all the above cases we can show that

$$\max \mathcal{A}_t^f - \max \hat{\mathcal{A}}_t^f \leq \frac{\langle X_t, \tilde{\mu} - \mu^* \rangle}{\tau}.$$

Let's prove this one by one.

### A.3.1  1st case

In this case, it is true that $\max \mathcal{A}_t^f - \max \hat{\mathcal{A}}_t^f = 1 - 1 = 0$.

### A.3.2  2nd case

The non-trivial pair in this case is $\left( 1, \dfrac{\tau}{\gamma_c \left( \sqrt{2 \log\left(\frac{1}{\delta}\right)} \right) + \langle X_t, \tilde{\mu} \rangle} \right)$.

When the above relation for $\max \mathcal{A}_t^f$ and $\max \hat{\mathcal{A}}_t^f$ holds, then it is true that:

1. $\gamma_c \left( \sqrt{2 \log\left(\frac{1}{\delta}\right)} \right) + \langle X_t, \mu^* \rangle \leq 0$,

2. $\dfrac{\tau}{\gamma_c \left( \sqrt{2 \log\left(\frac{1}{\delta}\right)} \right) + \langle X_t, \tilde{\mu} \rangle} \leq 1$.

Using the above, we can bound $1 - \dfrac{\tau}{\gamma_c \left( \sqrt{2 \log\left(\frac{1}{\delta}\right)} \right) + \langle X_t, \tilde{\mu} \rangle}$ as follows:

$$
\begin{aligned}
1 - \frac{\tau}{\gamma_c \left( \sqrt{2 \log\left(\frac{1}{\delta}\right)} \right) + \langle X_t, \tilde{\mu} \rangle} &= \frac{\gamma_c \left( \sqrt{2 \log\left(\frac{1}{\delta}\right)} \right) - \tau + \langle X_t, \tilde{\mu} \rangle}{\gamma_c \left( \sqrt{2 \log\left(\frac{1}{\delta}\right)} \right) + \langle X_t, \tilde{\mu} \rangle} \\
&\leq \frac{-\langle X_t, \mu^* \rangle + \langle X_t, \tilde{\mu} \rangle}{\tau} \\
&= \frac{\langle X_t, \tilde{\mu} - \mu^* \rangle}{\tau}.
\end{aligned}
$$

### A.3.3  3rd case

We choose $\tilde{\mu}$ pessimistically, so in this case the only valid pair is $(1, 1)$ and $\mid \tilde{\alpha}_t - \alpha_t \mid = 0$.

### A.3.4  4th case

This case can be divided into four different subcases:

1. $(1, 1)$ then $\mid \tilde{\alpha}_t - \alpha_t \mid = 0$.

2. $\left( 1, \dfrac{\tau}{\gamma_c \left( \sqrt{2 \log\left(\frac{1}{\delta}\right)} \right) + \langle X_t, \tilde{\mu} \rangle} \right)$. We saw that the above case can be bounded by $\dfrac{\langle X_t, \tilde{\mu} - \mu^* \rangle}{\tau}$.

3. $\left( \dfrac{\tau}{\gamma_c \left( \sqrt{2 \log\left(\frac{1}{\delta}\right)} \right) + \langle X_t, \mu^* \rangle}, 1 \right)$; this case cannot exist due to the way we choose $\tilde{\mu}$.

4. $\left( \dfrac{\tau}{\gamma_c \left( \sqrt{2 \log\left(\frac{1}{\delta}\right)} \right) + \langle X_t, \mu^* \rangle}, \dfrac{\tau}{\gamma_c \left( \sqrt{2 \log\left(\frac{1}{\delta}\right)} \right) + \langle X_t, \tilde{\mu} \rangle} \right)$.

In this case, it holds that $0 < \tau < \gamma_c \left( \sqrt{2 \log\left(\frac{1}{\delta}\right)} \right) + \langle X_t, \mu^* \rangle$ and $0 < \tau < \gamma_c \left( \sqrt{2 \log\left(\frac{1}{\delta}\right)} \right) + \langle X_t, \tilde{\mu} \rangle$.

We are going to bound $|\max \mathcal{A}_t^f - \max \hat{\mathcal{A}}_t^f|$ as follows:

$$|\max \mathcal{A}_t^f - \max \hat{\mathcal{A}}_t^f| = \left| \frac{\tau}{\gamma_c \left(\sqrt{2\log\left(\frac{1}{\delta}\right)}\right) + \langle X_t, \mu^* \rangle} - \frac{\tau}{\gamma_c \left(\sqrt{2\log\left(\frac{1}{\delta}\right)}\right) + \langle X_t, \tilde{\mu} \rangle} \right|$$

$$= \left| \frac{\tau \langle X_t, \tilde{\mu} - \mu^* \rangle}{\left(\gamma_c \left(\sqrt{2\log\left(\frac{1}{\delta}\right)}\right) + \langle X_t, \mu^* \rangle\right)\left(\gamma_c \left(\sqrt{2\log\left(\frac{1}{\delta}\right)}\right) + \langle X_t, \tilde{\mu} \rangle\right)} \right|$$

$$\leq \frac{\tau \langle X_t, \tilde{\mu} - \mu^* \rangle|}{\tau^2}$$

$$= \frac{\langle X_t, \tilde{\mu} - \mu^* \rangle}{\tau}.$$

### A.4  Proof of theorem 2

*Proof.*

$$\mathcal{R}_{\mathcal{C}}(T) = \sum_{t=1}^{T}(\alpha_t^* - \tilde{\alpha}_t)\langle X_t, \theta^* \rangle + \sum_{t=1}^{T}(\tilde{\alpha}_t - \alpha_t)\langle X_t, \theta^* \rangle$$

$$\leq \sum_{t=1}^{T} |\tilde{\alpha}_t| \, \|x_t\|_{\Sigma_t^{-1}} \left\|\tilde{\theta} - \theta^*\right\|_{\Sigma_t} + \frac{LS}{\tau} \sum_{t=1}^{T} |\tilde{\alpha}_t| \, \|x_t\|_{\Sigma_t^{-1}} \|\tilde{\mu} - \mu^*\|_{\Sigma_t}$$

$$\leq \sum_{t=1}^{T} \beta_t(\delta', d) \, \|x_t\|_{\Sigma_t^{-1}} + \frac{LS}{\tau} \sum_{t=1}^{T} \beta_t(\delta', d) \, \|x_t\|_{\Sigma_t^{-1}}$$

$$\leq \beta_T(\delta', d)(1 + \frac{LS}{\tau}) \left( \sum_{t=1}^{T} \|x_t\|_{\Sigma_t^{-1}} \right)$$

$$\leq \beta_T(\delta', d)(1 + \frac{LS}{\tau}) \sqrt{2Td\log\left(1 + \frac{TL^2}{\lambda}\right)}$$

$\square$

## B  Non-Linear case

### B.1  Bound of $\alpha_t^* \theta_*(X_t) - \tilde{\alpha}_t \theta_*(X_t)$

*Proof.* The proof of the lemma 4 is similar to [23] as the decision set is the same for both $\alpha_t^*$ and $\tilde{\alpha}_t$. We define $U_t(\alpha) = \sup\{\alpha\theta_*(X_t) : \theta_* \in \mathcal{G}_r\}$ and $L_t(\alpha) = \inf\{\alpha\theta_*(X_t) : \theta_* \in \mathcal{G}_r\}$. When $\theta_*$ lies in $\mathcal{G}_r$ it holds that $L_t(\alpha) \leq \theta_*(\alpha) \leq U_t(\alpha)$. Using this we derive

$$\alpha_t^* \theta_*(X_t) - \tilde{\alpha}_t \theta_*(X_t) \leq (U_t(\alpha_t^*) - L_t(\tilde{\alpha}_t))\,\mathbf{1}\{\}\theta_* \in \mathcal{G}_r) + 2\mathbf{1}\{\}\theta_* \notin \mathcal{G}_r)$$
$$\leq (U_t(\alpha_t^*) - L_t(\tilde{\alpha}_t)) + 2\mathbf{1}\{\}\theta_* \notin \mathcal{G}_r)$$
$$\leq w_{\mathcal{G}_r}(X_t) + 2\mathbf{1}\{\}\theta_* \notin \mathcal{G}_r) + \underbrace{[U_t(\alpha_t^*) - U_t(\tilde{\alpha}_t)]}_{\leq 0 \text{ due to selection rule}}$$

(17)

$\square$

Where in the last line we also used the fact that $\tilde{\alpha} \in [0, 1]$.

### B.2  Analyzing the cost for approximating $\mu_*(X_t)$

We need to bound $|\tilde{\alpha}_t - \alpha_t|$. First, we remind the definitions of $\tilde{\alpha}_t$ and $\alpha_t$:

$$\tilde{\alpha}_t = \arg\max_{\alpha \in \mathcal{A}_t^f}\{\alpha\hat{\theta}_*(X_t)\},$$

$$\alpha_t = \arg\max_{\alpha \in \hat{\mathcal{A}}_t^f}\{\alpha\hat{\theta}_*(X_t)\}.$$

374 We observe that both the choice of $\tilde{\alpha}_t$ and the choice of $\alpha_t$ depend on the sign of the value of $\hat{\theta}_*(X_t)$.

375 If $\hat{\theta}_*(X_t) \geq 0$, then $\tilde{\alpha}_t$ equals the maximum element of the set $\mathcal{A}_t^f$. Similarly, $\alpha_t$ equals the maximum

376 of the set $\hat{\mathcal{A}}_t^f$ when $\hat{\theta}_*(X_t) \geq 0$. On the other side, when $\hat{\theta}_*(X_t) < 0$, both $\tilde{\alpha}_t$ and $\alpha_t$ are zero.

We will write down again the sets $\mathcal{A}_t^f$ and $\hat{\mathcal{A}}_t^f$ to see the possible values for $(\tilde{\alpha}_t, \alpha_t)$:

$$\mathcal{A}_t^f = \left\{ \alpha \in [0,1] : \left( \mu_*(X_t) + \gamma_c \left( \sqrt{2\log\left(\frac{1}{\delta}\right)} \right) \right) \alpha \leq \tau \right\},$$

$$\hat{\mathcal{A}}_t^f = \left\{ \alpha \in [0,1] : \left( \tilde{\mu}(X_t) + \gamma_c \left( \sqrt{2\log\left(\frac{1}{\delta}\right)} \right) \right) \alpha \leq \tau \right\}.$$

377 Our estimator $\tilde{\mu}$ for $\mu^*$ is a pessimistic one. Among all possible choices for $\tilde{\mu}$, in order to be robust,

378 we will choose $\tilde{\mu}$ such that $\hat{\mathcal{A}}_t^f$ has the smallest possible length.

379 Having that in mind, we have the four following scenarios for $(\tilde{\alpha}_t, \alpha_t)$:

380    1. $(1,1)$

381    2. $\left( 1, \min\left( 1, \dfrac{\tau}{\gamma_c \left( \sqrt{2\log\left(\frac{1}{\delta}\right)} \right) + \tilde{\mu}(X_t)} \right) \right)$

382    3. $\left( \min\left( 1, \dfrac{\tau}{\gamma_c \left( \sqrt{2\log\left(\frac{1}{\delta}\right)} \right) + \mu_*(X_t)} \right), 1 \right)$

383    4. $\left( \min\left( 1, \dfrac{\tau}{\gamma_c \left( \sqrt{2\log\left(\frac{1}{\delta}\right)} \right) + \mu_*(X_t)} \right), \min\left( 1, \dfrac{\tau}{\gamma_c \left( \sqrt{2\log\left(\frac{1}{\delta}\right)} \right) + \tilde{\mu}(X_t)} \right) \right)$

384 For all the above cases we can show that

$$\max \mathcal{A}_t^f - \max \hat{\mathcal{A}}_t^f \leq \frac{\tilde{\mu}(X_t) - \mu_*(X_t)}{\tau}.$$

385 Let's prove this one by one.

### B.2.1   1st case

387 In this case, it is true that $\max \mathcal{A}_t^f - \max \hat{\mathcal{A}}_t^f = 1 - 1 = 0$.

### B.2.2   2nd case

389 The non-trivial pair in this case is $\left( 1, \dfrac{\tau}{\gamma_c \left( \sqrt{2\log\left(\frac{1}{\delta}\right)} \right) + \tilde{\mu}(X_t)} \right)$.

390 When the above relation for $\max \mathcal{A}_t^f$ and $\max \hat{\mathcal{A}}_t^f$ holds, then it is true that:

391    1. $\gamma_c \left( \sqrt{2\log\left(\frac{1}{\delta}\right)} \right) + \mu_*(X_t) \leq 0$,

392    2. $\dfrac{\tau}{\gamma_c \left( \sqrt{2\log\left(\frac{1}{\delta}\right)} \right) + \tilde{\mu}(X_t)} \leq 1$.

Using the above, we can bound $1 - \dfrac{\tau}{\gamma_c \left( \sqrt{2 \log \left( \frac{1}{\delta} \right)} \right) + \tilde{\mu}(X_t)}$ as follows:

$$1 - \frac{\tau}{\gamma_c \left( \sqrt{2 \log \left( \frac{1}{\delta} \right)} \right) + \tilde{\mu}(X_t)} = \frac{\gamma_c \left( \sqrt{2 \log \left( \frac{1}{\delta} \right)} \right) - \tau + \tilde{\mu}(X_t)}{\gamma_c \left( \sqrt{2 \log \left( \frac{1}{\delta} \right)} \right) + \tilde{\mu}(X_t)}$$
$$\leq \frac{-\mu_*(X_t) + \tilde{\mu}(X_t)}{\tau}$$
$$= \frac{\tilde{\mu}(X_t) - \mu_*(X_t)}{\tau}.$$

### B.2.3  3rd case

We choose $\tilde{\mu}$ pessimistically, so in this case the only valid pair is $(1, 1)$ and $\mid \tilde{\alpha}_t - \alpha_t \mid = 0$.

### B.2.4  4th case

This case can be divided into four different subcases:

1. $(1, 1)$ then $\mid \tilde{\alpha}_t - \alpha_t \mid = 0$.

2. $\left( 1, \dfrac{\tau}{\gamma_c \left( \sqrt{2 \log \left( \frac{1}{\delta} \right)} \right) + \tilde{\mu}(X_t)} \right)$.  We saw that the above case can be bounded by $\dfrac{\tilde{\mu}(X_t) - \mu_*(X_t)}{\tau}$.

3. $\left( \dfrac{\tau}{\gamma_c \left( \sqrt{2 \log \left( \frac{1}{\delta} \right)} \right) + \mu_*(X_t)}, 1 \right)$; this case cannot exist due to the way we choose $\tilde{\mu}$.

4. $\left( \dfrac{\tau}{\gamma_c \left( \sqrt{2 \log \left( \frac{1}{\delta} \right)} \right) + \mu_*(X_t)}, \dfrac{\tau}{\gamma_c \left( \sqrt{2 \log \left( \frac{1}{\delta} \right)} \right) + \tilde{\mu}(X_t)} \right)$.

In this case, it holds that $0 < \tau < \gamma_c \left( \sqrt{2 \log \left( \frac{1}{\delta} \right)} \right) + \mu_*(X_t)$ and $0 < \tau < \gamma_c \left( \sqrt{2 \log \left( \frac{1}{\delta} \right)} \right) + \tilde{\mu}(X_t)$.

We are going to bound $\mid \max \mathcal{A}_t^f - \max \hat{\mathcal{A}}_t^f \mid$ as follows:

$$\mid \max \mathcal{A}_t^f - \max \hat{\mathcal{A}}_t^f \mid = \left| \frac{\tau}{\gamma_c \left( \sqrt{2 \log \left( \frac{1}{\delta} \right)} \right) + \mu_*(X_t)} - \frac{\tau}{\gamma_c \left( \sqrt{2 \log \left( \frac{1}{\delta} \right)} \right) + \tilde{\mu}(X_t)} \right|$$
$$= \left| \frac{\tau \left( \langle X_t, \tilde{\mu} \rangle - \mu_*(X_t) \right)}{\left( \gamma_c \left( \sqrt{2 \log \left( \frac{1}{\delta} \right)} \right) + \mu_*(X_t) \right) \left( \gamma_c \left( \sqrt{2 \log \left( \frac{1}{\delta} \right)} \right) + \tilde{\mu}(X_t) \right)} \right|$$
$$\leq \frac{\tau \mid \tilde{\mu}(X_t) - \mu_*(X_t) \mid}{\tau^2}$$
$$= \frac{\tilde{\mu}(X_t) - \mu_*(X_t)}{\tau}.$$

Now by following exaclty the same procedure as in lemma 4 we derive that $|\tilde{\alpha}_t - \alpha_t| \leq w_{\mathcal{G}_c}(A)/\tau$.

## C  Experimental Results

As mentioned earlier, potential applications of this problem include advertising, optimal dosage determination, and reinforcement learning from human feedback (RLHF). However, obtaining

suitable data to evaluate the algorithm is challenging for the first two applications, while the last is left for future exploration. Consequently, in this initial version of our work, we evaluate the algorithm using synthetic data.

To produce $\theta^*$ and $\mu^*$, these entities were drawn from a $d$-dimensional normal distribution, followed by normalization. Similarly, the contexts were derived from a multivariate normal distribution and subsequently normalized. The experiments were conducted employing vectors of 5 and 10 dimensions, utilizing various values of $\tau$ across $5 \times 10^4$ iterations. In practical terms, it is pertinent to explore the interrelations between $\tau$ and $\max \|X\|$, $\|\theta^*\|$, and $\|\mu^*\|$, as these are intrinsically linked to the problem's formulation, feature selection, and the choice of $\tau$.

Our observations indicate that for larger values of $\tau$, such as those exceeding $0.5$, there is an increase in regret. This phenomenon is anticipated since a lower threshold constrains the algorithm more significantly, thereby facilitating a more rapid exploration of the available dosage space. Furthermore, it was observed that for larger values of $\tau$, including $0.6$ and $0.8$, the results exhibited a sub-linear progression after $10^4$ iterations. Notably, after $4 \times 10^5$ iterations, we detected a stabilization in growth, suggesting the convergence of our estimators to the true values of $\theta^*$ and $\mu^*$, accompanied by reduced confidence intervals.

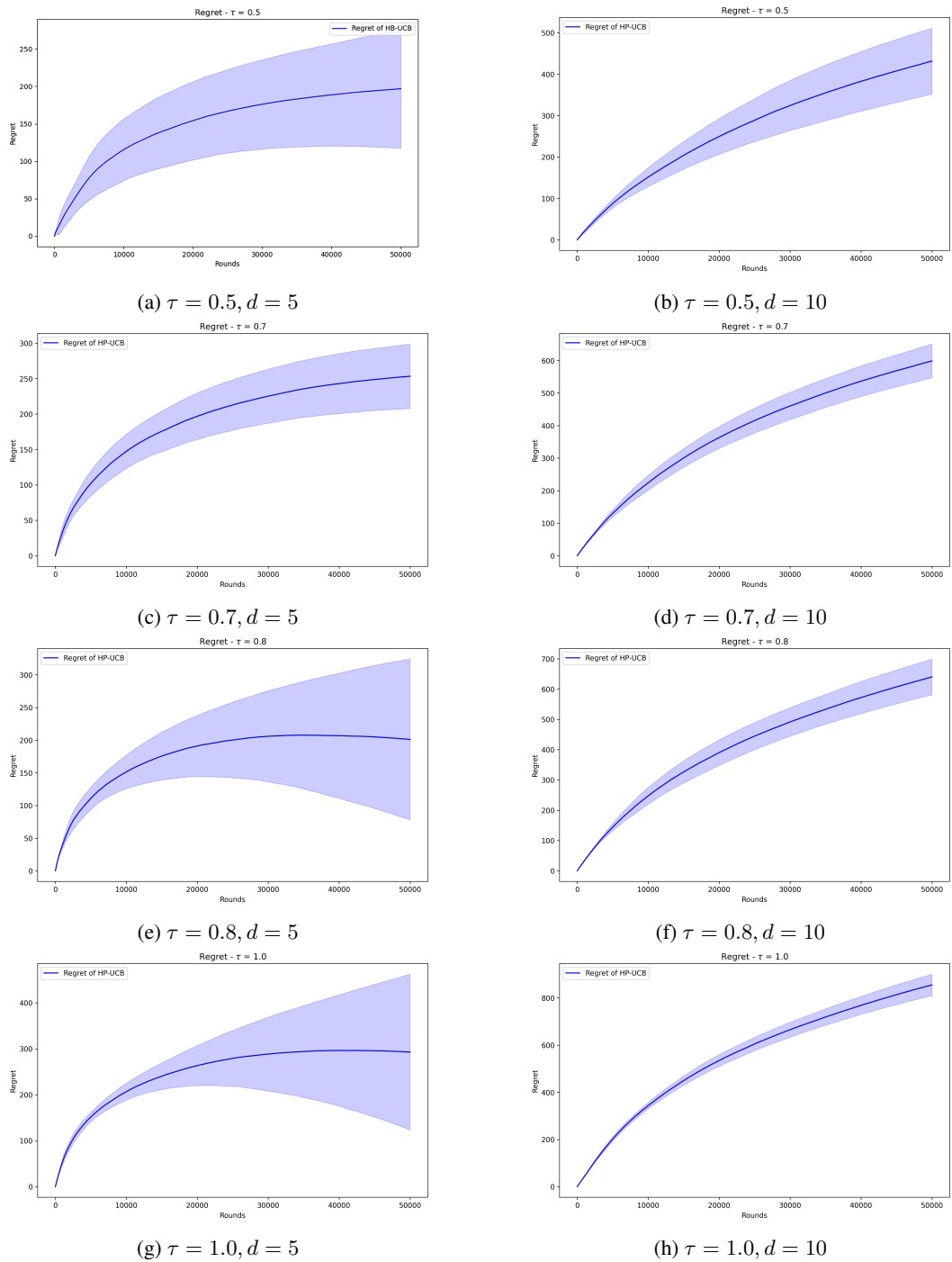

Figure 1: Plots of the regret for various $\tau$ and $d$ values.

