# OpenReview forum: "Safety by Design: High-Probability Constrained Contextual Bandits"
_NeurIPS.cc/2025/Workshop/Reliable_ML — NeurIPS 2025 - Reliable ML Workshop_

### Official Review · Reviewer_Ep91 · 2025-09-13
**A theoretical paper**

**Rating:** 5
**Confidence:** 1

**Review:**

**Summary.**
This paper aims to enforce safety constraints in multi-armed bandit algorithms. Existing works mainly focus on ensuring that the expected value of the cost signal remains below a desired threshold, with constraints satisfied with high probability. In contrast, this paper proposes a new UCB-style algorithm to make actual cost at each step to stay within a safe region with high probability.

**Strengths.**
The proposed algorithm makes the actual cost at each step to stay within a safe region with high probability, not just on average.

**Weaknesses / Limitations.**
It is a bit hard to follow the paper. It would be appreciated if the authors could clarify the preliminaries and related work with more details.

The experiments in this paper are done mainly on synthetic data, rather than real-world data. The practical deployment in medicine or RLHF will need more evaluation.

**Ethics.**
None.